# Structures and Properties of 4-phpy, pyz, and 4,4′-bpy Adducts of Lantern-Type Dirhodium Complexes with µ-Formamidinato and µ-Carboxylato Bridges

**Makoto Handa [1,\*]**, **Satoshi Nishiura [1]**, **Makoto Kano [1]**, **Natsumi Yano [1]**, **Haruo Akashi [2]**, **Masahiro Mikuriya [3]**, **Hidekazu Tanaka [1]**, **Tatsuya Kawamoto [4]** and **Yusuke Kataoka [1]**

[1] Department of Chemistry, Graduate School of Natural Science and Technology, Shimane University, 1060 Nishikawatsu, Matsue 690-8504, Japan; sa.to.nishi421@gmail.com (S.N.); hananona.ppm.917@gmail.com (M.K.); s179802@matsu.shimane-u.ac.jp (N.Y.); hidekazu@riko.shimane-u.ac.jp (H.T.); kataoka@riko.shimane-u.ac.jp (Y.K.)

[2] Institute of Frontier Science and Technology, Okayama University of Science, 1-1 Ridaicho, Kita-Ku, Okayama 700-0005, Japan; akashi@ifst.ous.ac.jp

[3] Department of Applied Chemistry for Environment, School of Science and Technology, Kwansei Gakuin University, 2-1 Gakuen, Sanda 669-1337, Japan; junpei@kwansei.ac.jp

[4] Department of Chemistry, Faculty of Science, Kanagawa University, 2946 Tsuchiya, Hiratsuka, Kanagawa 259-1293, Japan; kaw@kanagawa-u.ac.jp

\* Correspondence: handam@riko.shimane-u.ac.jp; Tel.: +81-852-32-6418

**Abstract:** Dinuclear and polymer complexes of 4-phenylpyridine (4-phpy), pyazine (pyz), and 4,4′-bipyridine (4,4′-bpy) were prepared by using *cis*-[Rh$_2$(4-Me-pf)$_2$(O$_2$CR)$_2$] (4-Me-pf$^-$ =$N,N'$-bis(4-methylphenyl)formamidinate anion; R = CF$_3$ and CMe$_3$) as precursor dinuclear units. The dinuclear structures of *cis*-[Rh$_2$$^{II,II}$(4-Me-pf)$_2$(O$_2$CR)$_2$(4-phpy)$_2$] and *cis*-[Rh$_2$$^{II,III}$(4-Me-pf)$_2$(O$_2$CCMe$_3$)$_2$(4-phpy)$_2$]BF$_4$ and polymer structures of [Rh$_2$$^{II,II}$(4-Me-pf)$_2$(O$_2$CR)$_2$(L)]$_n$ (L = pyz and 4,4′-bpy) were confirmed by X-ray crystal structure analyses. In these complexes, the lantern-type dinuclear core structures with *cis*-(2:2) arrangement of formamidinato (4-Me-pf$^-$) and carboxylato ligands are preserved with Rh–Rh distances of 2.44–2.47 Å, regardless of the difference in the axial ligand and oxidation state Rh$_2$$^{II,II}$ or Rh$_2$$^{II,III}$. In the cyclic voltammograms (CVs) in CH$_2$Cl$_2$, the redox potentials for Rh$_2$$^{II,III}$/Rh$_2$$^{II,II}$ were estimated as $E_{1/2}$ = 0.07 V and $-0.28$ V (vs. Fc$^+$/Fc) for *cis*-[Rh$_2$(4-Me-pf)$_2$(O$_2$CCF$_3$)$_2$(4-phpy)$_2$] and *cis*-[Rh$_2$(4-Me-pf)$_2$(O$_2$CCMe$_3$)$_2$(4-phpy)$_2$], respectively, negatively shifted by 0.16 and 0.12 V compared with those of corresponding parent dinuclear complexes. The results were interpreted that the axial interaction with 4-phpy ligands makes the Rh$_2$$^{II,II}$ core oxidized easily. The oxidized complex *cis*-[Rh$_2$(4-Me-pf)$_2$(O$_2$CCMe$_3$)$_2$(4-phpy)$_2$]BF$_4$ is paramagnetic, which was confirmed by effective magnetic moment value $\mu_{eff}$ = 1.90 µ$_B$ at 300 K per Rh$_2$$^{II,III}$ unit ($S$ = 1/2). The polymer complexes [Rh$_2$(4-Me-pf)$_2$(O$_2$CR)$_2$(L)]$_n$ (L = pyz and 4,4′-bpy) showed Type II gas-adsorption properties for N$_2$.

**Keywords:** lantern-type dirhodium complex; formamidinate; trifluoroacetate; pivalate; mixed-ligand complexes; bis-adduct complexes; polymer complexes

## 1. Introduction

Tetra-µ-carboxylatodimetal complexes ([M$_2$(O$_2$CR)$_4$]$^{0/n+}$) with a lantern-type structure have been known for many transition metal ions [1,2]. We have been studying on the use of the dimetal complexes as building blocks in combination with $N,N'$-donating linker ligands such as pyrazine (pyz) and 4,4′-bipyridine (4,4′-bpy) to produce assembled polymer complexes [3–24]. We have reported N$_2$ gas occlusion properties of a pyz-linked complex [Cu$_2$(O$_2$CPh)$_4$(pyz)]$_n$ [9,10]. The CO$_2$ gas occlusion was later reported for a polymer complex of tetrabenzoatodirhodium(II) units linked by pyrazine [Rh$_2$(O$_2$CPh)$_4$(pyz)]$_n$ [25]. $N,N'$-bis(alkylphenyl)formamidinate anions (R-pf$^-$) work as bidentate ligands to form the

lantern-type dinuclear complexes [Rh$_2$(R-pf)$_4$] (R-pf$^-$ = *N,N'*-bis(alkylphenyl)formamidinate anion (Scheme 1)) [1,2]. In 1987, Piraino reported on synthesis and a crystal structure of [Rh$_2$(4-Me-pf)$_4$] (4-Me-pf$^-$ =*N,N'*-bis(4- methylphenyl)formamidinate anion), which was synthesized through *cis*-[Rh$_2$(4-Me-pf)(O$_2$CCF$_3$)$_2$] (**1**) (Scheme 2a) by using its reaction with H(4-Me-pf) [26,27].

**Scheme 1.** Chemical structure of 4-R-pf $^-$.

**(a)** *cis*-[Rh$_2$(4-Me-pf)$_2$(O$_2$CCF$_3$)$_2$] (**1**)　　　　**(b)** *cis*-[Rh$_2$(4-Me-pf)$_2$(O$_2$CCMe$_3$)$_2$] (**2**)

**Scheme 2.** Chemical structures of *cis*-[Rh$_2$(4-Me-pf)$_2$(O$_2$CR)$_2$] (R = CF$_3$ (**1**) and CMe$_3$(**2**)).

The formamidinate ion more strongly coordinates to metal ion than carboxylate ion because the amidinate nitrogen has a stronger donating nature than the carboxylate oxygen. The robust dinuclear complexes [Rh$_2$(R-pf)$_4$] are considered to be suitable as the building block in combination with linker ligands for producing their assembled complexes. However, both of the axial sites of [Rh$_2$(R-pf)$_4$] are crowded with four aryl groups of the amidinato-bridges, which makes the coordination of axial linkers such as pyz and 4,4'-bpy ligands impossible. Only a bidentate ligand 1,4-diisocyanobenzene (1,4-dib) with longer NC ligating arms from a benzene ring (Scheme 3) can coordinate to the Rh ion through the axial crowded space, giving a polymer complex [Rh$_2$(4-Me-pf)$_4$(1,4-dib)]$_n$, which was isolated from the benzene and toluene solution [28,29]. A hexanuclear complex [{Rh$_2$(4-Me-pf)$_4$}$_3$(1,4-dib)$_2$] was isolated from the dichloromethane solution [28], while a tetranuclear complex [{Rh$_2$(dpf)$_4$}$_2$(1,4-dib)] (dpf$^-$ = *N,N'*-diphenylformamidinate anion) was also isolated from dichloromethane solution [30]. We preliminarily showed that pyz could work as a linker ligand for *cis*-[Rh$_2$(R-pf)$_2$(O$_2$CCF$_3$)$_2$] with less crowded axial sites compared with [Rh$_2$(R-pf)$_4$] based on the crystal structure of [Rh$_2$(4-Et-pf)$_2$(O$_2$CCF$_3$)$_2$(pyz)]$_n$·*n*(toluene) (4-Et-pf$^-$ = *N,N'*-bis(4-ethylphenyl)formamidinate anion), such a study has not been performed since then [31].

**Scheme 3.** Chemical structure of 1,4-dib.

We recently reported on the crystal structure and absorption spectral and electrochemical properties of *cis*-[Rh$_2$(4-Me-pf)$_2$(O$_2$CCMe$_3$)$_2$] (**2**) (Scheme 2b), which was synthesized by the substitution of trifluoroacetato (CF$_3$CO$_2^-$) bridges of **1** to pivalato (CMe$_3$CO$_2^-$) bridges [32]. Complex **2** is relatively easy to be oxidized from Rh$_2^{II,II}$ to Rh$_2^{II,III}$ species; $E_{1/2}$ for Rh$_2^{II,III}$/Rh$_2^{II,II}$ (measured by cyclic voltammograms (CVs) in CH$_2$Cl$_2$ containing 0.1 M TBA(PF$_6$)) = 0.32 V vs SCE (for **2**), 0.71 V vs SCE (for **1**), and 0.09 V vs. SCE (for [Rh$_2$(4-Me-pf)$_4$]) [27]. This is advantageous as a building block giving the paramagnetic assembly based on the spin ($S$ = 1/2) of the Rh$_2^{II,III}$ species, whereas the Rh$_2^{II,II}$ species is diamagnetic. Such assembled Rh$_2^{II,III}$ complexes are hopefully expected as new magnetic materials showing gas occlusion properties. We started the research to examine the following three points: (1) Less crowded axial sites of **1** and **2** can be coordinated by bidentate linker ligands such as pyz and 4,4′-bpy, (2) Rh$_2^{II,III}$ complexes with the axial ligands can be isolated without decomposition and are characterized by magnetic and absorption spectral measurements to clarify their potentialities as building blocks, (3) Molecular or assembled structures are determined by X-ray single-crystal structure analysis, which is one of the most powerful characterization tools to see if the desired compounds are formed. We successfully obtained bis-adduct Rh$_2^{II,II}$ and Rh$_2^{II,III}$ complexes of 4-phpy (4-phenylpyridine) [Rh$_2^{II,II}$(4-Me-pf)$_2$(O$_2$CR)$_2$(4-phpy)$_2$] (R = CF$_3$ and CMe$_3$) and [Rh$_2^{II,III}$(4-Me-pf)$_2$(O$_2$CR)$_2$(4-phpy)$_2$]BF$_4$ and polymer complexes [Rh$_2^{II,II}$(4-Me-pf)$_2$(O$_2$CR)$_2$(L)]$_n$ (L = pyz and 4,4′-bpy). Their crystal structures, spectral, electrochemical and magnetic properties and N$_2$ gas adsorption properties are presented in this report.

## 2. Results and Discussion

### 2.1. Bis-Adduct Rh$_2^{II,II}$ Complexes of 4-phpy

The molecular structures of *cis*-[Rh$_2$(4-Me-pf)$_2$(O$_2$CCF$_3$)$_2$(4-phpy)$_2$] (**3**) and *cis*-[Rh$_2$(4-Me-pf)$_2$(O$_2$CCMe$_3$)$_2$(4-phpy)$_2$] (**4**) are shown in Figures 1 and 2, respectively, which confirm the *cis*-(2:2) arrangement of the bridging ligands. The 4-phpy ligand is coordinated to each dinuclear unit with distances of 2.3045(17) Å for **3** and 2.289(2) Å for **4**, respectively. The Rh–Rh bond distances are 2.4702(3) Å for **3** and 2.4428(4) Å for **4**, respectively. The Rh–Rh distances of dinuclear complexes *cis*-[Rh$_2$(4-Me-pf)$_2$(O$_2$CCF$_3$)$_2$(H$_2$O)$_2$]·0.5C$_6$H$_6$ (**1**(H$_2$O)$_2$·0.5C$_6$H$_6$), *cis*-[Rh$_2$(4-Me-pf)$_2$(O$_2$CCF$_3$)$_2$(MeCN)$_2$] (**1**(MeCN)$_2$) and *cis*-[Rh$_2$(4-Me-pf)$_2$(O$_2$CCMe$_3$)$_2$(MeCN)$_2$] (**2**(MeCN)$_2$) were reported to be 2.425(1) Å, 2.474(5) Å, and 2.4343(6) Å, respectively [26,32,33]. The enhanced acidity of Rh(II) ion due to the introduced fluorine atoms on carboxylate bridges of **1** could lead to large elongation of Rh–Rh distance found for **1**(MeCN)$_2$ (2.474(5) Å) and **3** (2.4702(3) Å), when taking into account that **1**(H$_2$O)$_2$•0.5C$_6$H$_6$ has a shorter Rh–Rh distance of 2.425(1) Å with axial coordination of less basic water oxygen atoms [26]. The dinuclear complexes **2**(MeCN)$_2$ and **4** have similar Rh–Rh distances of 2.4343(6) Å for **2**(MeCN)$_2$ and 2.4428(4) Å for **4**, which means that MeCN and 4-phpy have similar basicity for the Rh atom of the dinuclear complexes. Packing diagram of **4** is shown in Figure 3. There is a short contact between carbon atoms, which are described by dotted lines, probably due to the π–π stacking (C•••C = 3.343 Å).

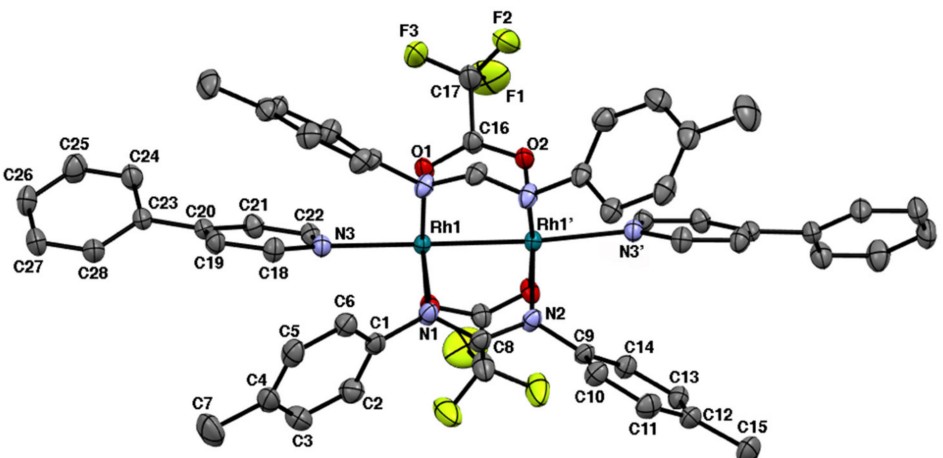

**Figure 1.** ORTEP view of *cis*-[Rh$_2$(4-Me-pf)$_2$(O$_2$CCF$_3$)$_2$(4-phpy)$_2$] (**3**), showing thermal ellipsoids at the 50% probability level. Hydrogen atoms are omitted for clarity. The prime refers to the equivalent position (1/2–x, 1/2–y, z).

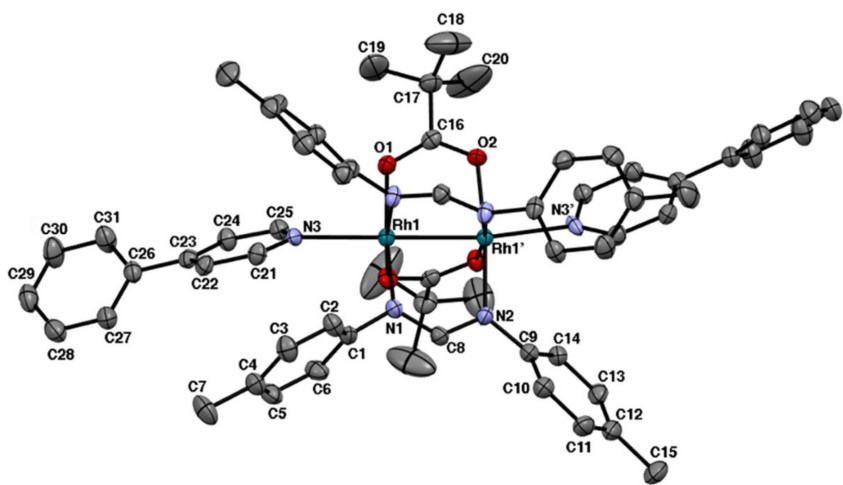

**Figure 2.** ORTEP view of *cis*-[Rh$_2$(4-Me-pf)$_2$(O$_2$CCMe$_3$)$_2$(4-phpy)$_2$] (**4**), showing thermal ellipsoids at the 50% probability level. Hydrogen atoms are omitted for clarity. The prime refers to the equivalent position (1–x, y, 1/2–z).

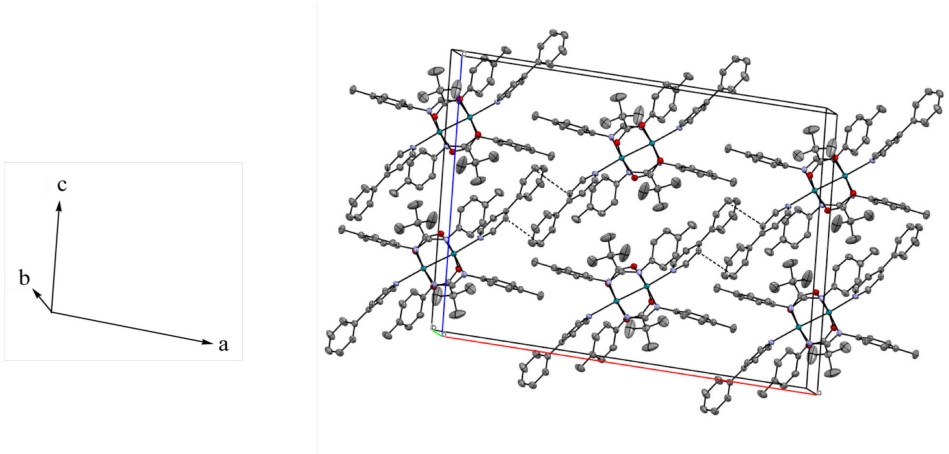

**Figure 3.** Packing diagram of *cis*-[Rh$_2$(4-Me-pf)$_2$(O$_2$CCMe$_3$)$_2$(4-phpy)$_2$] (**4**).

Diffuse reflectance spectra of **3** and **4** are shown in Figure 4, with those of respective parent dinuclear complexes of **1** and **2**. On the axial coordination of 4-phpy, spectral features are changed for both of the complexes **3** and **4**, compared with those of the parent dinuclear complexes. Usually, the visible bands (600 ~ 800 nm) assigned as $\pi^* \rightarrow \sigma^*$ transition within the $Rh_2$ core for lantern-type dirhodium(II) tetracarboxylates are blue-shifted due to the axial $\sigma$-interaction, which makes $\sigma^*$ orbital energy higher [34,35]. Such axial interactions may occur in **3** and **4**; the visible bands at $\lambda_{max}$ = 592 nm for **1** and $\lambda_{max}$ = 670 nm for **2** are blue-shifted to $\lambda_{max}$ = 515 nm for **3** and $\lambda_{max}$ = 500 nm for **4**, respectively. However, the visible band origins of the present complexes could be different from the $\pi^* \rightarrow \sigma^*$ transitions within the dinuclear cores because, based on the time-dependent density functional theory (TD-DFT) calculation results, the main component of the band observed in the longest wavelength region for **2** ($\lambda_{max}$ = 613 nm in MeCN) was previously assigned as a $\delta^*$ (HOMO) $\rightarrow \sigma^*$ (LUMO) transition within the dinuclear core, although the corresponding lowest energy band of **1** ($\lambda_{max}$ = 516 nm in MeCN) was also assigned as a $\delta^*$ (HOMO) $\rightarrow \delta^*$ (LUMO + 1) transition [32]. The axial interactions were also confirmed to be present in the $CH_2Cl_2$ solutions. Absorption spectra of **1–4** measured in $CH_2Cl_2$ with $1.0 \times 10^{-4}$ M are given in Figure 5. The visible bands of **3** ($\lambda_{max}$ = 571 nm) and **4** ($\lambda_{max}$ = 584 nm) appear at shorter wavelength region compared with those of **1** ($\lambda_{max}$ = 675 nm) and **2** ($\lambda_{max}$ = 700 nm), respectively. The band positions for **3** and **4** are nearly the same at higher concentration ($1.0 \times 10^{-3}$ M), respectively (Figure S1).

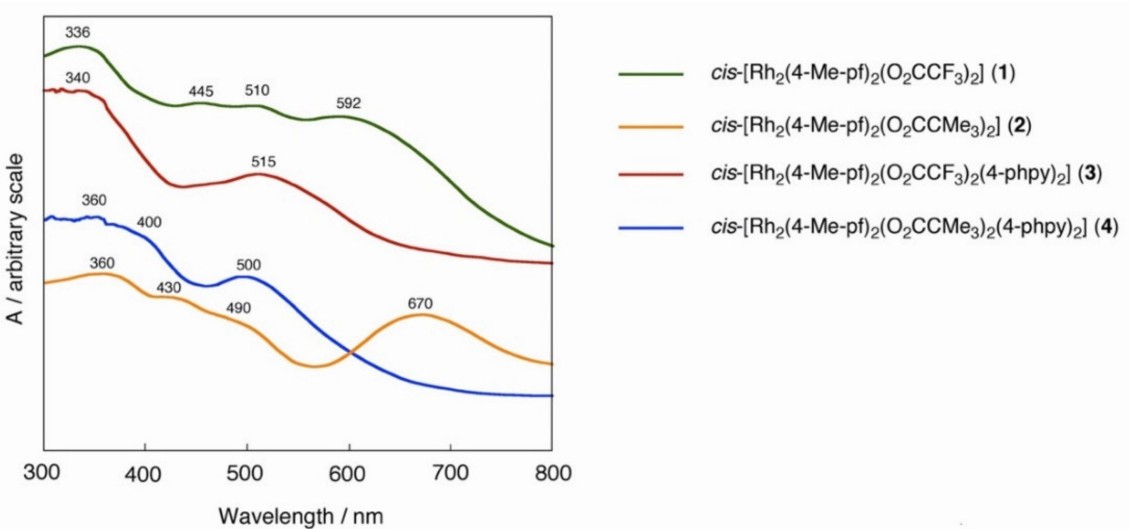

**Figure 4.** Diffuse reflectance spectra of *cis*-[$Rh_2$(4-Mepf)$_2$($O_2CCF_3$)$_2$(4-phpy)$_2$] (**3**) and *cis*-[$Rh_2$(4-Me-pf)$_2$($O_2CCMe_3$)$_2$(4-phpy)$_2$] (**4**) with their parent dinuclear complexes *cis*-[$Rh_2$(4-Me-pf)$_2$($O_2CCF_3$)$_2$] (**1**) and *cis*-[$Rh_2$(4-Me-pf)$_2$($O_2CCMe_3$)$_2$] (**2**).

Cyclic voltammograms (CVs) were measured for **3** and **4** in $CH_2Cl_2$ containing 0.1 M TBA(PF$_6$) (Figure 6). The redox waves for $Rh_2^{II,II} \rightarrow Rh_2^{II,III}$ process were observed at $E_{1/2}$ ($(E_{pa} + E_{pc})/2$) = 0.07 V (vs. Fc$^+$/Fc) for **3** and −0.28 V (vs. Fc$^+$/Fc) for **4**, where Fc means ferrocene. On taking into account that the Fc$^+$/Fc values has been reported as $E_{1/2}$ = 0.48 V (vs. SCE) in $CH_2Cl_2$ [36], the redox values for $Rh_2^{II,II} \rightarrow Rh_2^{II,III}$ process were estimated as 0.55 V for **3** and 0.20 V for **4** relative to SCE. It is shown that **3** and **4** are easily oxidized compared with their parent dinuclear complexes **1** ($E_{1/2}$ = 0.71 V (vs. SCE) in $CH_2Cl_2$) and **2** ($E_{1/2}$ = 0.32 V (vs. SCE) in $CH_2Cl_2$), which may be due to the axial interaction with 4-phpy. It should be also noticed that strong $\sigma$-donations coming from formamidinato ligands make the formamidinato-bridged $Rh_2^{II,II}$ complexes oxidized easily, compared with corresponding dirhodium(II) tetracarboxylate dinuclear complexes, e.g., $E_{1/2}$ (for [$Rh_2^{II,III}$($O_2CCMe_3$)$_4$]$^+$/[$Rh_2^{II,III}$($O_2CCMe_3$)$_4$]$^+$) = 1.12 V (vs. SCE) in $CH_2Cl_2$, although $E_{1/2}$ (for [$Rh_2^{II,III}$(4-Me-pf)$_4$]$^+$/[$Rh_2^{II,II}$(4-Me-pf)$_4$]) = 0.09 V (vs. SCE) in $CH_2Cl_2$ [32]. The difference in the redox potential $E_{1/2}$ ($Rh_2^{II,III}$/$Rh_2^{II,II}$) between **3** (0.55 V vs. SCE) and

**4** (0.20 V vs. SCE) is due to the difference in the substituent group on the carboxylato bridge between electron-withdrawing trifluoromethyl groups introduced for **3** and electron-donating *t*-butyl groups for **4**, as has been explained for the parent dinuclear complexes **1** ($E_{1/2}$ (for **1**$^+$/**1**) = 0.71 V (vs. SCE) in CH$_2$Cl$_2$ and **2** ($E_{1/2}$ (for **2**$^+$/**2**) = 0.32 V (vs. SCE) in CH$_2$Cl$_2$ [32].

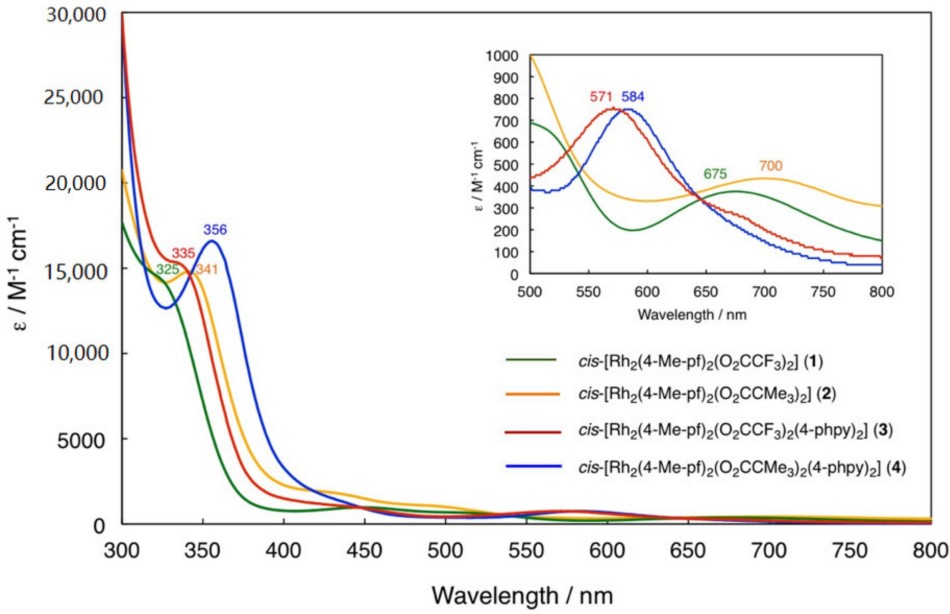

**Figure 5.** Absorption spectra of *cis*-[Rh$_2$(4-Mepf)$_2$(O$_2$CCF$_3$)$_2$(4-phpy)$_2$] (**3**) and *cis*-[Rh$_2$(4-Me-pf)$_2$(O$_2$CCMe$_3$)$_2$(4-phpy)$_2$] (**4**) with their parent dinuclear complexes *cis*-[Rh$_2$(4-Me-pf)$_2$(O$_2$CCF$_3$)$_2$] (**1**) and *cis*-[Rh$_2$(4-Me-pf)$_2$(O$_2$CCMe$_3$)$_2$] (**2**) in CH$_2$Cl$_2$ at the concentration of $1 \times 10^{-4}$ M.

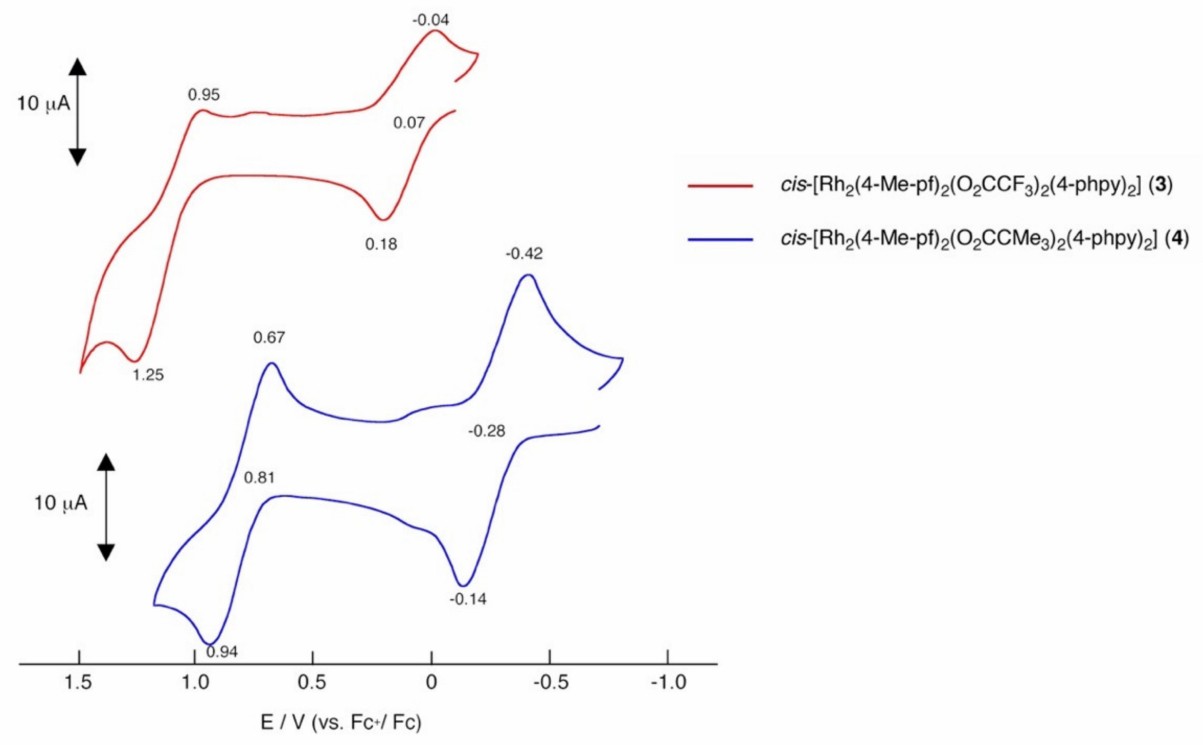

**Figure 6.** Cyclic voltammograms of *cis*-[Rh$_2$(4-Me-pf)$_2$(O$_2$CCF$_3$)$_2$(4-phpy)$_2$] (**3**) and *cis*-[Rh$_2$(4-Me-pf)$_2$(O$_2$CCMe$_3$)$_2$(4-phpy)$_2$] (**4**) at $1.0 \times 10^{-3}$ M in CH$_2$Cl$_2$ containing 0.1 M TBA(PF$_6$) (Glassy carbon working electrode; san rate = 50 mV/s).

## 2.2. Bis-Adduct $Rh_2^{II,III}$ Complexes of 4-phpy

As has been described in the previous section, the dinuclear complex **4** is relatively easy to oxidize. We have succeeded in isolating single crystals of *cis*-[$Rh_2^{II,III}$(4-Me-pf)$_2$(O$_2$CCMe$_3$)$_2$(4-phpy)$_2$]BF$_4$·2MeOH (**5**) suitable for X-ray crystal structure analysis. The cationic dinuclear part of **5** is shown in Figure 7, which confirms the *cis*-(2:2) arrangement of the bridging ligands. The crystal solvent molecules of MeOH exist among *cis*-[Rh$_2$(4-Me-pf)$_2$(O$_2$CCMe$_3$)$_2$(4-phpy)$_2$]$^+$ cations and BF$_4^-$ anions as shown in the packing diagram (Figure 8). One of two crystallographically independent MeOH molecules has a disorder at the oxygen atom and hence it is divided with the occupancy weights of 0.6 and 0.4. The axial ligand 4-phpy is coordinated to Rh atoms with distances of 2.2673(16) Å (for Rh1-N5) and 2.2935(16) Å (for Rh2-N6), which is comparable to that of **4** (2.289(2) Å). The Rh1-Rh2 bond distances are 2.45320(19) Å, which is also comparable to that of **4** (2.4428(4) Å). It seems to be strange, because the bond order of Rh$_2$ core increases from 1 to 1.5 on the oxidation when an electron is removed from the antibonding $\pi^*$ orbital; the electronic structure of the lantern-type $Rh_2^{II,II}$ complexes are generally known as $\sigma^2\pi^4\delta^2\delta^{*2}\pi^{*4}$ [1,2]. In fact, the Rh–Rh bond distance of [$Rh_2^{II,III}$(O$_2$CMe)$_4$(H$_2$O)$_2$]ClO$_4$·H$_2$O (2.315(2) and 2.318(2) Å) [37] are shorter than that of [$Rh_2^{II,II}$(O$_2$CMe)$_4$(H$_2$O)$_2$] (2.386(1) Å) [38,39]. However, this trend is not clearly shown when the bridging ligand contains *N* donor atom. The Rh–Rh bond distances of [$Rh_2^{II,III}$(HNOCMe)$_4$(H$_2$O)$_2$]ClO$_4$ [40], [$Rh_2^{II,III}$(HNOCMe)$_4$(H$_2$O)$_2$]PF$_6$ [41], [$Rh_2^{II,III}$(HNOCMe)$_4$(H$_2$O)$_2$]PF$_6$· 2H$_2$O [41], and [$Rh_2^{II,III}$(HNOCMe)$_4$(H$_2$O)$_2$][ReO$_4$] [42] are 2.399(1) Å, 2.4084(10), 2.4015(4), and 2.4053(7), respectively, while that of [$Rh_2^{II,II}$(HNOCMe)$_4$(H$_2$O)$_2$]· 3H$_2$O is 2.415(1) Å [43]. The Rh–Rh bond distance of $Rh_2^{II,III}$ dinuclear complex [Rh$_2$(dpf)$_4^{II,III}$(MeCN)$_2$]ClO$_4$ is 2.466(1) Å, while that of corresponding $Rh_2^{II,II}$ dinuclear complex [$Rh_2^{II,II}$(dpf)$_4$(MeCN)$_2$] is 2.459(1) Å [44]. The Rh–Rh distance of [$Rh_2^{II,III}$(4-Me-pf)$_4$]ClO$_4$ is 2.447(1) Å [45] and that of [Rh$_2$(4-Me-pf)$_4$] is 2.4336(4) Å [27]. These results might be related to stronger σ-donation of amidato or amidinato nitrogen compared with carboxylato oxygen. We also reported that axial py coordinated complex [$Rh_2^{II,III}$(HNOCMe)$_4$(py)$_2$]BF$_4$ has a longer Rh–Rh bond (2.434(1) Å) relative to that of [$Rh_2^{II,III}$(HNOCMe)$_4$(H$_2$O)$_2$]ClO$_4$ (2.399(1) Å) [46]. The Rh–Rh bond distances of the related complexes discussed above are summarized in Table S1.

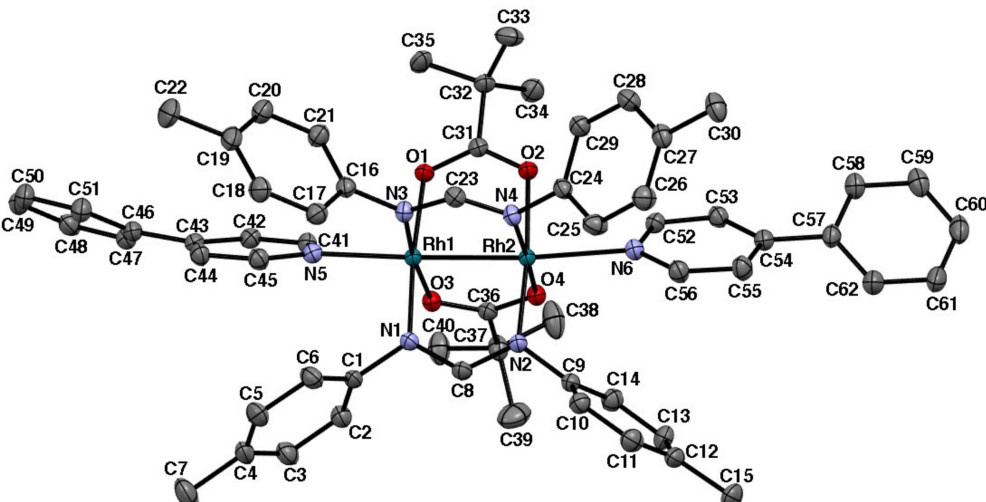

**Figure 7.** ORTEP view of cationic $Rh_2^{II,III}$ dinuclear core of *cis*-[Rh$_2$(4-Mepf)$_2$(O$_2$CCMe$_3$)$_2$(4-phpy)$_2$]BF$_4$·2MeOH (**5**·2MeOH), showing thermal ellipsoids at the 50% probability level. Hydrogen atoms are omitted for clarity.

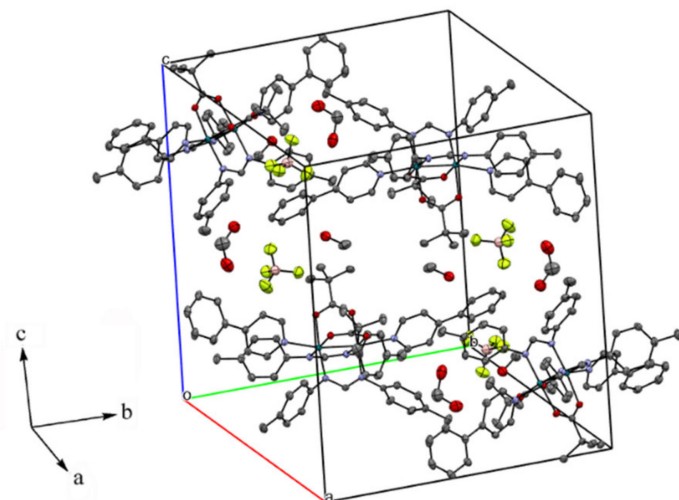

**Figure 8.** Packing diagram of *cis*-[Rh$_2$(4-Mepf)$_2$(O$_2$CCMe$_3$)$_2$(4-phpy)$_2$]BF$_4$·2MeOH (**5**·2MeOH).

Diffuse reflectance spectrum of **5** is given in Figure 9. The broad band around 1100 nm is characteristic of the mixed-valence cationic Rh$_2$$^{II,III}$ species [44]. In order to examine the existence of an unpaired electron within the dinuclear Rh$_2$$^{II,III}$ core of **5**, magnetic susceptibility measurement was performed. The effective magnetic moment at 300 K is 1.90 μ$_B$ per Rh$_2$ unit, indicative of the existence of an unpaired electron with $S = 1/2$. The temperature-dependent reciprocal magnetic susceptibilities over 2–300 K temperature range and the field-dependent magnetization from 0 to 70000 G at 2.0 K are shown in Figures 10 and 11, respectively. $\chi_M$$^{-1}$ obeys the Curie–Weiss law, $\chi_M = C/(T - \theta)$ with $C = 0.477(4)$ cm$^3$ mol$^{-1}$ K and $\theta = -14(2)$ K, suggesting that the antiferromagnetic interaction between Rh$_2$$^{II,III}$ units is limited and weakly operative. This may be because there is no considerable short contact among the Rh$_2$$^{II,III}$ units, as shown in the packing diagram (Figure 8).

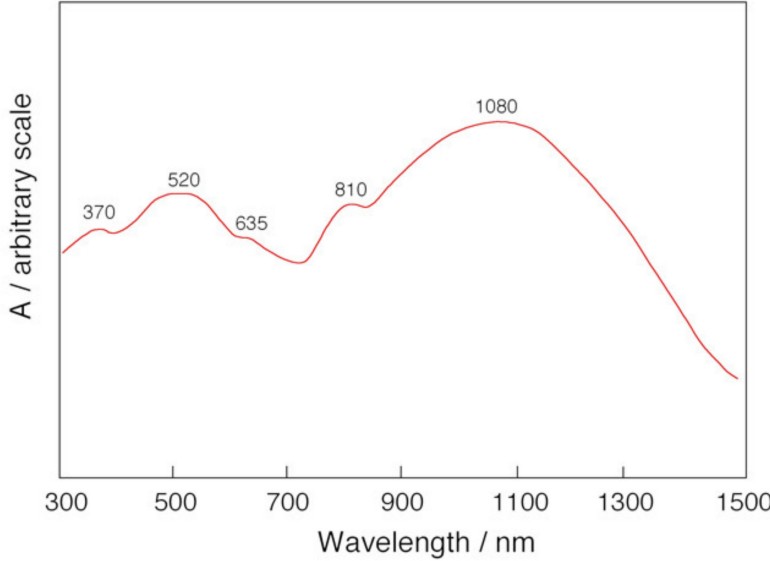

**Figure 9.** Diffuse reflectance spectrum of *cis*-[Rh$_2$(4-M-epf)$_2$(O$_2$CCMe$_3$)$_2$(4-phpy)$_2$]BF$_4$ (**5**).

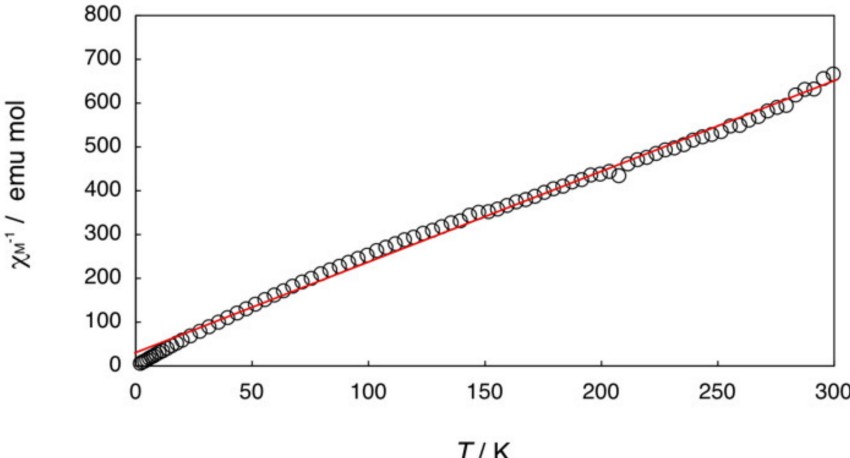

**Figure 10.** Temperature-dependent reciprocal magnetic susceptibilities ($\chi_M{}^{-1}$) of *cis*-[Rh$_2$(4-Me-pf)$_2$(O$_2$CCMe$_3$)$_2$(4-phpy)$_2$]BF$_4$ (**5**).

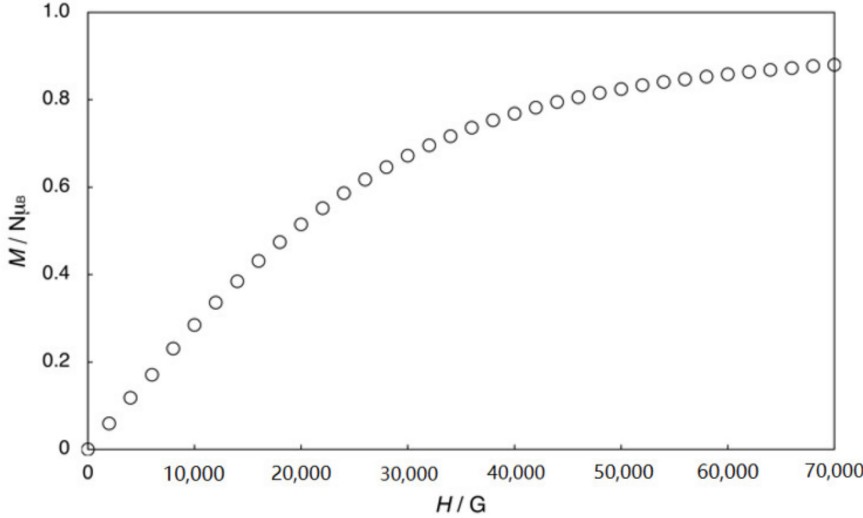

**Figure 11.** Field dependence plots of magnetization at 2.0 K for *cis*-[Rh$_2$(4-Me-pf)$_2$(O$_2$CCMe$_3$)$_2$(4-phpy)$_2$]BF$_4$ (**5**).

### 2.3. Rh$_2{}^{II,II}$ Polymer Complexes of pyz and 4,4′-bpy

The X-ray crystallographically determined polymer structures of [Rh$_2$(4-Me-pf)$_2$ (O$_2$CCF$_3$)$_2$(pyz)]$_n$ (**6**), [Rh$_2$(4-Me-pf)$_2$(O$_2$CCMe$_3$)$_2$(pyz)]$_n$ (**7**), [Rh$_2$(4-Me-pf)$_4$(O$_2$CCF$_3$)$_2$(4,4′-bpy)]$_n$ (**8**), and [Rh$_2$(4-Me-pf)$_2$(O$_2$CCMe$_3$)$_2$(4,4′-bpy)] (**9**) are depicted in Figures 12–15, respectively. The Rh$_2{}^{II,II}$ dinuclear core units are axially coordinated by nitrogen atoms of pyz or 4,4′-bpy bidentate linker ligand with distances of Rh1-N5 and Rh2-N6 of 2.031(4) and 2.320(4) Å for **6**, 2.302(4) and 2.283(4) Å for **7**, 2.300(6) and 2.294(6) Å for **8**, and 2.303(3) and 2.297(3) Å for **9**, respectively, to give polymer structures with an alternated arrangement of Rh$_2{}^{II,II}$ dinuclear cores and axial ligand liker units. The Rh1-Rh2 distances are 2.464(3) for **6**, 2.4421(5) Å for **7**, 2.4678(9) Å for **8**, and 2.4660(7) Å for **9**, respectively. Further, the Rh–Rh distance of [Rh$_2$(4-Et-pf)$_2$(O$_2$CCF$_3$)$_2$(pyz)]$_n$·$n$(toluene) is 2.4564(10) Å [31]. These Rh–Rh bond distances are not out of the range shown for the bis-adduct complexes **3** (2.4702(3) Å), **4** (2.4428(4) Å), and **5**·2MeOH (2.45320(19) Å). In Table 1, Rh–Rh, Rh-N$_{eq}$ (4-Me-pf), Rh-O$_{eq}$ (O$_2$CR), and Rh-N$_{ax}$ (axial ligands) are summarized for the present complexes.

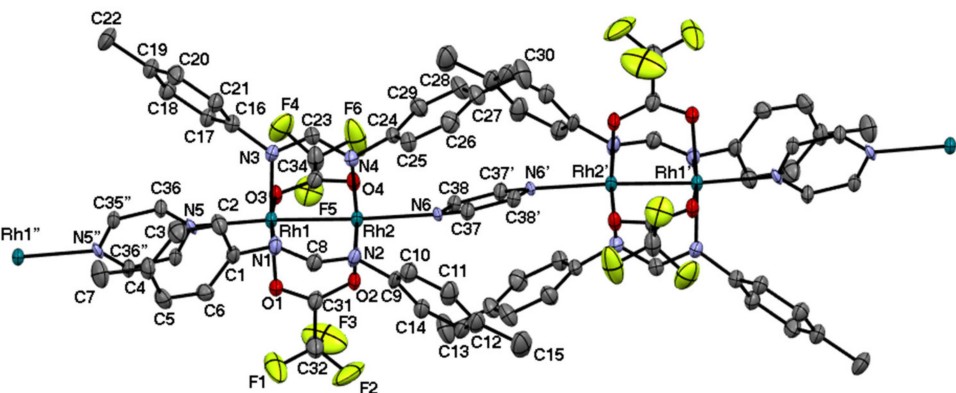

**Figure 12.** ORTEP view of [Rh$_2$(4-Me-pf)$_2$(O$_2$CCF$_3$)$_2$(pyz)]$_n$ (**6**), showing thermal ellipsoids at the 50% probability level. Hydrogen atoms are omitted for clarity. The prime and double prime refers to the equivalent positions (1–x, 1–y, 1-z) and (2–x, 2-y, 2–z), respectively.

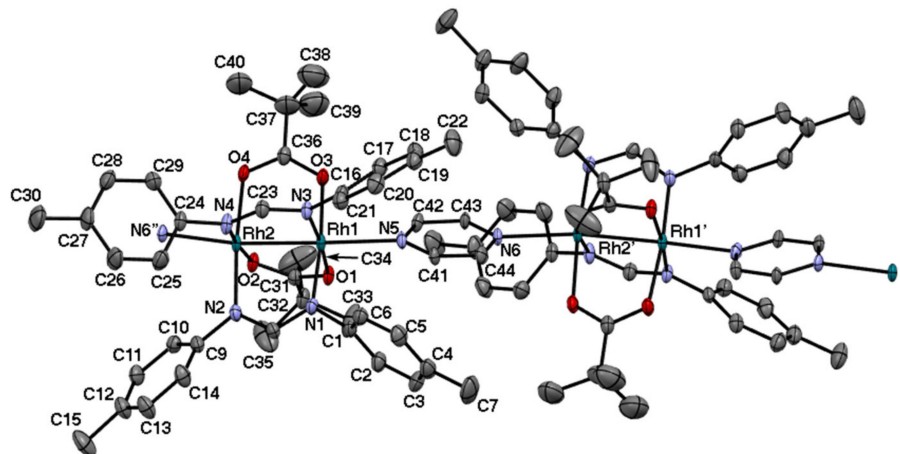

**Figure 13.** ORTEP view of [Rh$_2$(4-Me-pf)$_2$(O$_2$CCMe$_3$)$_2$(pyz)]$_n$ (**7**), showing thermal ellipsoids at the 50% probability level. Hydrogen atoms are omitted for clarity. The prime and double prime refer to the equivalent positions (–x, –1/2+y, 1/2-z) and (–x, 1/2+y, 1/2–z), respectively.

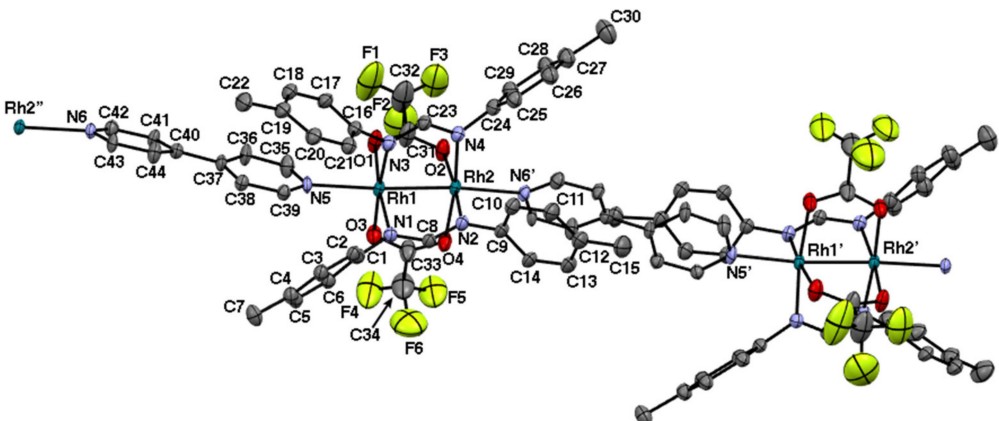

**Figure 14.** ORTEP view of [Rh$_2$(4-Me-pf)$_2$(O$_2$CCF$_3$)$_2$(4,4′-bpy)]$_n$ (**8**), showing thermal ellipsoids at the 50% probability level. Hydrogen atoms are omitted for clarity. The prime and double prime refer to the equivalent positions (–1+x, 1/2–y, –1/2+z) and (–1+x, 1/2–y, 1/2+z), respectively.

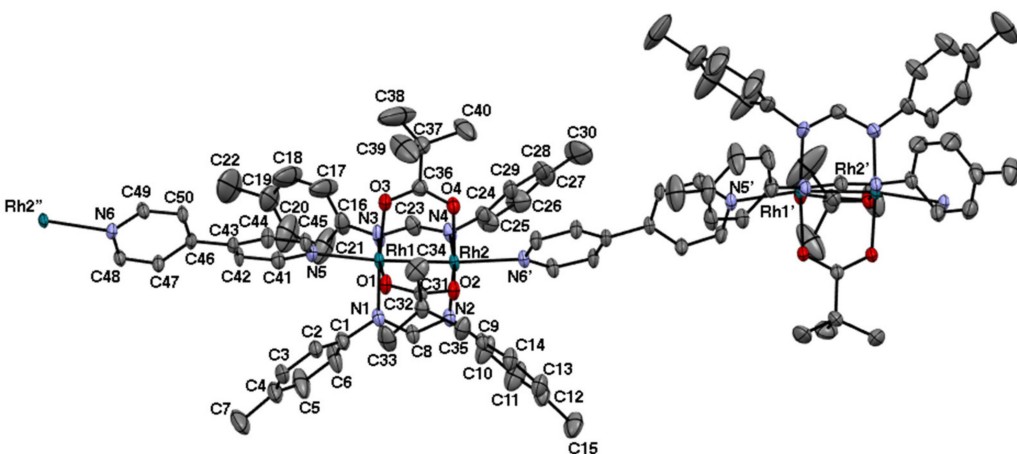

**Figure 15.** ORTEP view of [Rh$_2$(4-Me-pf)$_2$(O$_2$CCMe$_3$)$_2$(4,4′-bpy)]$_n$ (**9**), showing thermal ellipsoids at the 50% probability level. Hydrogen atoms are omitted for clarity. The prime and double prime refer to the equivalent positions (1/2+x, 1-y, z) and (−1/2+x, 1-y, z), respectively.

**Table 1.** Structural parameters of the present complexes **3**–**9**.

| Complexes | Rh–Rh/Å | Rh-N$_{eq}$ (4-Me-pf$^-$)/Å [a] | Rh-O$_{eq}$ (RCO$_2^-$)/Å [a] | Rh-N$_{ax}$/Å | T/K [b] |
|---|---|---|---|---|---|
| **3** | 2.4702(2) | 2.0069 | 2.0938 | 2.3045(17) | 150 |
| **4** | 2.4428(4) | 2.025 | 2.070 | 2.289(2) | 150 |
| **5•2MeOH** | 2.45320(19) | 1.9747 | 2.0487 | 2.2673(16), 2.2935(16) | 293 |
| **6** | 2.464(3) | 2.019 | 2.100 | 2.301(4), 2.320(4) | 150 |
| **7** | 2.4421(5) | 2.025 | 2.069 | 2.302(4), 2.283(4) | 150 |
| **8** | 2.4678(9) | 2.0193 | 2.102 | 2.300(6), 2.294(6) | 150 |
| **9** | 2.4660(7) | 2.023 | 2.066 | 2.303(3), 2.297(3) | 150 |

[a] Mean values; [b] measured temperatures.

It should be noted that the bis-adduct and adduct polymer complexes with equatorial CF$_3$CO$_2^-$ bridges **3**, **6** and **8** have rather elongated Rh–Rh bond distances, while the bis-adduct Rh$_2^{II,III}$ complex has relatively short Rh-N$_{eq}$ (4-Me-pf$^-$) and Rh-O$_{eq}$ (RCO$_2^-$) bonds, as has been similarly explained for the structural changes from Rh$_2^{II,II}$ complex [Rh$_2^{II,II}$(dpf)$_4$(MeCN)$_2$] to Rh$_2^{II,III}$ complex to [Rh$_2^{II,III}$(dpf)$_4$(MeCN)$_2$]ClO$_4$ [43]. It is interesting that the Rh$_2$ core adjust the core dimension with different manners for electron drawing fluorine group and higher oxidation state of Rh$_2^{II,III}$.

Diffuse reflectance spectra are shown for **1**, **6**, and **8** in Figure 16 and **2**, **7**, and **9** in Figure 17, respectively. The bands observed at 592 nm for **1** and 670 nm for **2** are both not found there on the formation of polymer complexes. This should be due to the axial interaction operative in the polymer complexes **6**–**9**, like the cases of bis-adduct Rh$_2^{II,II}$ complexes of 4-phpy **3** and **4**.

The adsorption isotherms of N$_2$ were measured at 77 K for the polymer complexes and the results are given in Figure 18 for **8** and Figures S2–S4 for **6**, **7**, and **9**, respectively. Their adsorption isotherms belong to Type II in IUPAC classification with a small $S_{BET}$ values of 2.2 m$^2$g$^{-1}$ for **6**, 4.4 m$^2$g$^{-1}$ for **7**, 39.5 m$^2$g$^{-1}$ for **8**, and 0.6 m$^2$g$^{-1}$ for **9**. A slight steep rise at a low relative pressure ($p/p_0$) was only recognizable for **8**. The void volumes per Rh$_2^{II,II}$, which were estimated by a PLATON SQUEEZE method (see Section 3.3 Crystal Structure Determination), are 253.0 Å$^3$ for **6**, 226.7 Å$^3$ for **7**, 113.3 Å$^3$ for **8**, and 244.8 Å$^3$ for

**9**. The void spaces for guest solvents in the single crystals are considered to collapse in the powder samples.

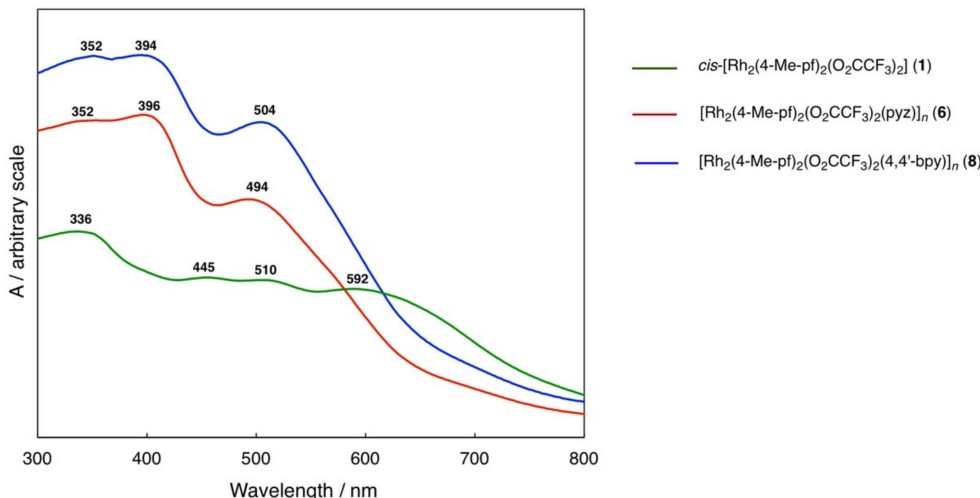

**Figure 16.** Diffuse reflectance spectra of *cis*-[Rh$_2$(4-Me-pf)$_2$(O$_2$CCF$_3$)$_2$] (**1**), [Rh$_2$(4-Me-pf)$_2$(O$_2$CCF$_3$)$_2$(pyz)]$_n$ (**6**), and [Rh$_2$(4-Me-pf)$_2$(O$_2$CCF$_3$)$_2$(4,4'-bpy)]$_n$ (**8**).

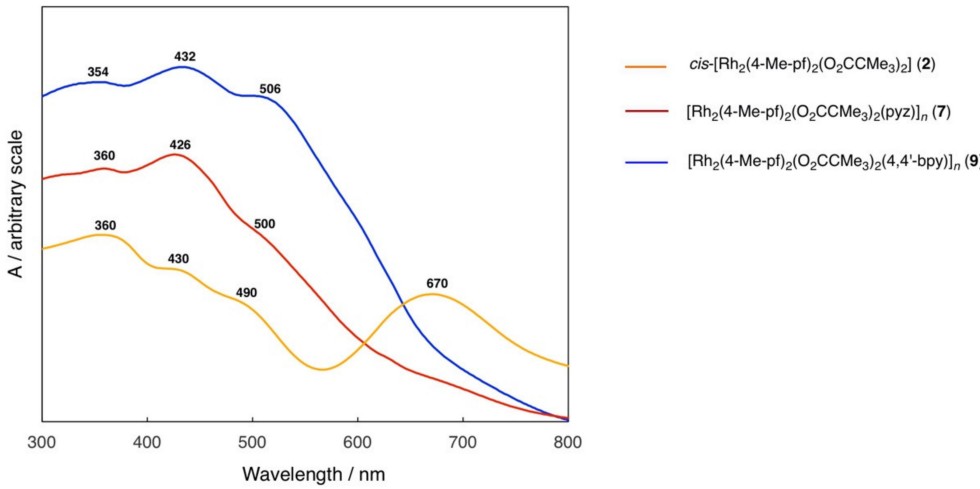

**Figure 17.** Diffuse reflectance spectra of *cis*-[Rh$_2$(4-Me-pf)$_2$(O$_2$CCMe$_3$)$_2$] (**2**), [Rh$_2$(4-Me-pf)$_2$(O$_2$CCMe$_3$)$_2$(pyz)]$_n$ (**7**), and [Rh$_2$(4-Me-pf)$_2$(O$_2$CCMe$_3$)$_2$(4,4'-bpy)]$_n$ (**9**).

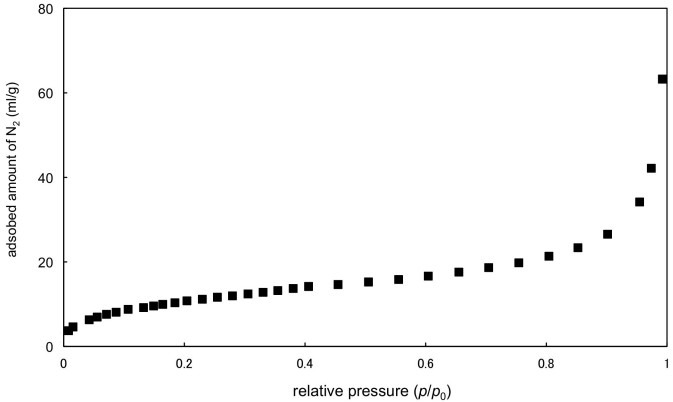

**Figure 18.** Nitrogen adsorption isotherm of [Rh$_2$(4-Me-pf)$_2$(O$_2$CCF$_3$)$_2$(4,4'-bpy)]$_n$ (**8**).

## 3. Materials and Methods

### 3.1. General Aspects

The formamidine ligand H(4-Me-pf) was obtained using the literature method [47]. The parent dinuclear complexes *cis*-[Rh$_2$(4-Me-pf)$_2$(O$_2$CCF$_3$)$_2$] (**1**) and *cis*-[Rh$_2$(4-Me-pf)$_2$(O$_2$CCMe$_3$)$_2$] (**2**) were prepared according to the methods described in the literatures [26,32].

Elemental analyses for carbon, hydrogen, and nitrogen were performed using a Yanako CHN Corder MT-6. Infrared spectra (KBr pellets) were measured with JASCO FT/IR-4600 and JASCO FT/IR-660 spectrometers for complexes **3**, **4** and **5–9**, respectively. Absorption spectra and diffuse reflectance spectra of **3** and **4** were obtained using a Shimadzu UV-2450 spectrometer. Diffuse reflectance spectra of **5–9** were obtained with a Shimadzu UV-3100 spectrometer. Cyclic voltammograms (CVs) were measured in dichloromethane containing tetra-*n*-butylammonium hexafluorophosphate (TBA(PF$_6$)) on a BAS 100BW Electrochemical Workstation. A glassy carbon disk (1.5 mm radius), platinum wire, and an Ag$^+$/Ag (tetra-*n*-butylammonium perchlorate/acetonitrile) electrodes were used as working, counter, and reference electrodes, respectively. Ferrocene (Fc) was used as an internal standard, and the potentials are quoted relative to Fc$^+$/Fc couple. The temperature dependence of magnetic susceptibilities over the temperature range of 2–300 K at the constant field of 5000 G and the field dependence of magnetization from 0 to 70,000 G at 2.0 K were measured with a Quantum Design MPMS 3. Adsorption measurements for N$_2$ was performed by a MicrotracBEL BELSOROP-mini II. Prior to the adsorptions, samples were evacuated at 298 K for 2 h.

### 3.2. Synthesis of Complexes

#### 3.2.1. Synthesis of *cis*-[Rh$_2$(4-Me-pf)$_2$(O$_2$CCF$_3$)$_2$(4-phpy)$_2$] (**3**)

The dinuclear complex **1** (89 mg, 0.10 mmol) and 4-phpy (40 mg, 0.26 mmol) were reacted in a mixed solvent of dichloromethane (10 mL) and methanol (10 mL) by stirring at room temperature overnight. The resultant precipitate was collected by suction and washed with water and dried by heating at 70 °C for 3 h under vacuum to give a reddish-pink powder. The yield was 58 mg (48% based on [Rh$_2$(4-Me-pf)$_2$(O$_2$CCF$_3$)$_2$]). Anal. found: C, 56.40, H, 4.18, N, 7.40. Calcd. for C$_{56}$H$_{48}$F$_6$N$_4$O$_4$Rh$_2$: C, 56.58, H, 4.07, N, 7.07%. IR data (KBr disk, cm$^{-1}$): 3024w (CH), 1662w (Ar), 1577s, 1505s, 1335w (COO), 1201s and 1150s (CF$_3$). Crystals suitable for X-ray crystal structure analysis were obtained by recrystallization from a solution of dichloromethane/methanol solvent mixture.

#### 3.2.2. Synthesis of *cis*-[Rh$_2$(4-Me-pf)$_2$(O$_2$CCMe$_3$)$_2$(4-phpy)$_2$] (**4**)

This complex was obtained as an orange powder by employing the same synthetic way as that of **3**, other than the use of complex **2** (85 mg, 0.10 mmol) as starting material instead of **1**. The yield was 103 mg (88% based on [Rh$_2$(4-Me-pf)$_2$(O$_2$CCMe$_3$)$_2$]). Anal. found: C, 63.63, H, 5.54, N, 7.50. Calcd. for C$_{62}$H$_{66}$N$_6$O$_4$Rh$_2$: C, 63.92, H, 5.71, N, 7.21%. IR data (KBr disk, cm$^{-1}$): 2954w (CH), 1622w (Ar), 1593s and 1505s (COO), 1416w (CH$_2$), 1335w (COO), 1225s (Ar). Crystals suitable for X-ray crystal structure analysis were obtained by recrystallization from a solution of dichloromethane/methanol solvent mixture.

#### 3.2.3. Synthesis of *cis*-[Rh$_2$(4-Me-pf)$_2$(O$_2$CCMe$_3$)$_2$(4-phpy)$_2$]BF$_4$ (**5**)

The starting Rh$_2^{II,III}$ complex *cis*-[Rh$_2$(4-Me-pf)$_2$(O$_2$CCMe$_3$)$_2$]BF$_4$ was prepared by oxidizing complex **2** (50 mg, 0.058 mmol) by nitrosonium tetrafluoroborate (NOBF$_4$) (10 mg, 0.086 mmol) with stirring in dichloromethane (25 mL) at room temperature for 4 h, followed by purifying by SiO$_2$ column with a CH$_2$Cl$_2$/MeOH (95:5 vol./vol.) solvent mixture as an eluent, evaporating to dryness, and heating at 80 °C under vacuum for 3 h. The yield was 41 mg (75% based on [Rh$_2$(4-Me-pf)$_2$(O$_2$CCMe$_3$)$_2$]). Anal. found: C, 50.90, H, 5.03, N, 6.03. Calcd. for C$_{40}$H$_{48}$BF$_4$N$_4$O$_4$Rh$_2$: C, 51.03, H, 5.14, N, 5.95%. IR data (KBr disk, cm$^{-1}$): 2959w (CH), 1642w, 1587w (Ar), 1503s, 1417s, 1377w (COO), 1216s (Ar), 1083s (BF$_4^-$).

The Rh$_2^{II,III}$ complex *cis*-[Rh$_2$(4-Me-pf)$_2$(O$_2$CCMe$_3$)$_2$]BF$_4$ (80 mg, 0.085 mmol) and 4-phpy (35 mg, 0.23 mmol) were mixed in a mixed solvent of dichloromethane (7 mL) and

methanol (10 mL) and left for 3 days without stirring to give reddish-brown crystals, which are single crystals suitable for X-ray crystal structure analysis. The crystals were collected by filtration, washed with water and methanol, and dried under vacuum at room temperature for 3 h. The yield was 29 mg (27% based on *cis*-[Rh$_2$(4-Me-pf)$_2$(O$_2$CCMe$_3$)$_2$]BF$_4$). Anal. found: C, 59.91, H, 5.07, N, 6.74. Calcd. for C$_{56}$H$_{48}$F$_6$N$_4$O$_4$Rh$_2$: C, 59.49, H, 5.31, N, 6.71%. IR data (KBr disk, cm$^{-1}$): 2957w (CH), 1607w (Ar), 1505s, 1418s (COO), 1213s (Ar), 1083s (BF$_4$).

### 3.2.4. Synthesis of *cis*-[Rh$_2$(4-Me-pf)$_2$(O$_2$CCF$_3$)$_2$(pyz)]$_n$ (**6**)

The dinuclear complex **1** (45 mg, 0.051 mmol) and pyz (10 mg, 0.13 mmol) were reacted in methanol (50 mL) by stirring at room temperature overnight. The resultant precipitate was collected by suction and washed with toluene and dried by heating at 80 °C for 5 h under vacuum to give a reddish-brown powder. The yield was 40 mg (82% based on [Rh$_2$(4-Me-pf)$_2$(O$_2$CCF$_3$)$_2$]. Anal. found: C, 47.44, H, 3.65, N, 8.56. Calcd. for C$_{38}$H$_{34}$F$_6$N$_6$O$_4$Rh$_2$: C, 47.62, H, 3.58, N, 8.77%. IR data (KBr disk, cm$^{-1}$): 3026w (CH), 1658m (Ar), 1578s (COO), 1506s (Ar), 1452w (COO), 1335w (CH$_2$), 1201s and 1156s (CF$_3$). Crystals suitable for X-ray crystal structure analysis were grown by the slow diffusion of pyz with complex **1** in chloroform, chlorobenzene, and toluene in a test tube.

### 3.2.5. Synthesis of *cis*-[Rh$_2$(4-Me-pf)$_2$(O$_2$CCMe$_3$)$_2$(pyz)]$_n$ (**7**)

The dinuclear complex **2** (50 mg, 0.058 mmol) and pyz (10 mg, 0.13 mmol) were reacted in methanol (50 mL) by stirring at room temperature for 5 h. The resultant precipitate was collected by suction and washed with ethanol, and dried by heating at 80 °C for 5 h under vacuum to give a yellowish-brown powder. The yield was 50 mg (91.2 % based on [Rh$_2$(4-Me-pf)$_2$(O$_2$CCMe$_3$)$_2$]). Anal. found: C, 56.09, H, 5.22, N, 8.47. Calcd. for C$_{44}$H$_{52}$N$_6$O$_4$Rh$_2$: C, 56.54, H, 5.61, N, 8.99%. IR data (KBr disk, cm$^{-1}$): 2957w (CH), 1621w (Ar), 1592s(COO) and 1504s (Ar), 1416m (COO), 1337w (CH$_2$), 1226s (Ar). Crystals suitable for X-ray crystal structure analysis were grown by the slow diffusion of 4,4′-bpy with complex **2** in toluene and methanol in a test tube.

### 3.2.6. Synthesis of *cis*-[Rh$_2$(4-Me-pf)$_2$(O$_2$CCF$_3$)$_2$(4,4′-bpy)]$_n$ (**8**)

This complex was obtained as a reddish-brown powder by employing the same synthetic way as that of **6**, other than the use of 4,4′-bpy (16 mg, 0.10 mmol) instead of pyz. The yield was 42.7 mg (81% based on [Rh$_2$(4-Me-pf)$_2$(O$_2$CCF$_3$)$_2$]). Anal. found: C, 50.91, H, 3.84, N, 7.84. Calcd. for C$_{44}$H$_{38}$F$_6$N$_6$O$_4$Rh$_2$: C, 51.08, H, 3.70, N, 8.12%. IR data (KBr disk, cm$^{-1}$): 3026w (CH), 1658m (Ar), 1578s (COO) and 1506s (Ar), 1452w (CH$_2$), 1335w (COO), 1201s and 1156s (CF$_3$). Crystals suitable for X-ray crystal structure analysis were grown by the slow diffusion of 4,4′-bpy with **1** in chloroform and methanol in a test tube.

### 3.2.7. Synthesis of *cis*-[Rh$_2$(4-Me-pf)$_2$(O$_2$CCMe$_3$)$_2$(4,4′-bpy)]$_n$ (**9**)

This complex was obtained as a brown powder by employing the same synthetic way as that of **7**, other than the use of 4,4′-bpy instead of pyz (complex **2** (46 mg, 0.054 mmol) and 4,4′-bpy (17 mg, 0.11 mmol) were used for the reaction). The yield was 38.7 mg (71% based on [Rh$_2$(4-Me-pf)$_2$(O$_2$CCMe$_3$)$_2$]). Anal. found: C, 59.59, H, 5.52, N, 8.27. Calcd. for C$_{50}$H$_{56}$N$_6$O$_4$Rh$_2$: C, 59.41, H, 5.58, N, 8.31%. IR data (KBr disk, cm$^{-1}$): 2958w (CH), 1624m (Ar), 1592s (COO) and 1505s (Ar), 1414w (CH$_2$), 1337w (COO), 1225w (Ar). Crystals suitable for X-ray crystal structure analysis were grown by the slow diffusion of 4,4′-bpy with complex **2** in dichloromethane and methanol in a test tube.

### *3.3. Crystal Structure Determination*

X-ray crystallographic data (Table 2) were collected for single crystals of **3** at 150 K on a RIGAKU HyPix6000 CCD system equipped with a Mo rotating-anode X-ray generator (λ = 0.71075 Å), installed at Institute for Molecular Science (IMS), for single crystals of **4** and **6** at 150 K on a Rigaku Mercury 70 CCD system equipped with a Mo rotating-anode

X-ray generator (λ = 0.71075 Å), installed at IMS, and for single crystals of **5** at 293 K and **7–9** at 150 K on a RIGAKU Saturn 724 CCD system equipped with a VariMax Mo rotating-anode X-ray generator (λ = 0.71075 Å), installed at Okayama University of Science and Kanagawa University, respectively. Diffraction data were processed using Crys- talClear and CrysAlisPro (RIGAKU). The structures were solved by direct methods (SIR) and refined using the full-matrix least-squares technique ($F^2$) with SHELXL-2018 as part of the CrystalStructure software (RIGAKU) for the complexes except for **5**, for which Olex2 was used [48]. Guest solvents of final refined models of **6–9** were excluded by PLATON SQUEEZE program because their solvents were heavily disordered. Non-hydrogen atoms were refined with anisotropic displacement parameters, and all hydrogen atoms were located at calculated positions and refined with a riding model. Selected bond distance and angles are given for **3–9** in Table S2.

**Table 2.** Crystallographic data and structure refinement of **3–9** [a].

| Complexes | 3 | 4 | 5·2MeOH | 6 | 7 | 8 | 9 |
|---|---|---|---|---|---|---|---|
| Empirical formula | $C_{56}H_{48}F_6N_6O_4Rh_2$ | $C_{62}H_{66}N_6O_4Rh_2$ | $C_{64}H_{70}BF_4N_6O_6Rh_2$ | $C_{38}H_{34}F_6N_6O_4Rh_2$ | $C_{44}H_{52}N_6O_4Rh_2$ | $C_{44}H_{38}F_6N_6O_4Rh_2$ | $C_{50}H_{56}N_6O_4Rh_2$ |
| Formula mass | 1188.82 | 1165.02 | 1311.89 | 958.53 | 934.73 | 1034.62 | 1010.82 |
| Temperature, $T$ (K) | 150 | 150 | 293 | 150 | 150 | 150 | 150 |
| Crystal system | orthorhombic | monoclinic | monoclinic | triclinic | monoclinic | monoclinic | monoclinic |
| Space group | $Pccn$ | $C2/c$ | $P2_1/n$ | $P1$ | $P2_1/c$ | $P2_1/c$ | $I2/a$ |
| $a$ (Å) | 14.3060(3) | 27.611(2) | 15.0200(2) | 11.517(14) | 10.8066(3) | 9.428(3) | 27.388(6) |
| $b$ (Å) | 17.9846(4) | 10.0319(8) | 20.7383(2) | 11.578(15) | 18.9112(4) | 21.162(6) | 10.390(2) |
| $c$ (Å) | 20.3568(6) | 20.3559(15) | 20.3334(2) | 18.91(2) | 24.2718(8) | 23.223(6) | 38.642(7) |
| $\alpha$ (°) | 90 | 90 | 90 | 95.639(15) | 90 | 90 | 90 |
| $\beta$ (°) | 90 | 94.966(2) | 107.2500(10) | 105.24(2) | 95.950(3) | 97.122(5) | 101.144(12) |
| $\gamma$ (°) | 90 | 90 | 90 | 108.23(3) | 90 | 90 | 90 |
| Unit-cell volume, $V$ (Å$^3$) | 5237.6(2) | 5617.2(7) | 6122.12(12) | 2266(5) | 4933.6(2) | 4598(2) | 10789(4) |
| Formula per unit cell, $Z$ | 4 | 4 | 4 | 2 | 4 | 4 | 8 |
| Density, $D_{calcd}$ (g cm$^{-3}$) | 1.508 | 1.378 | 1.423 | 1.405 | 1.258 | 1.495 | 1.245 |
| Crystal size (mm) | 0.602 × 0.399 × 0.103 | 0.350 × 0.200 × 0.170 | 0.20 × 0.10 × 0.050 | 0.100 × 0.100 × 0.010 | 0.220 × 0.020 × 0.020 | 0.130 × 0.100 × 0.030 | 0.220 × 0.200 × 0.190 |
| Absorption coefficient, μ (mm$^{-1}$) | 0.703 | 0.640 | 0.607 | 0.794 | 0.711 | 0.789 | 0.655 |
| $\theta$ range for data collection (°) | 1.819–25.999 | 2.161–24.997 | 1.481–31.594 | 2.500–26.368 | 2.634–24.248 | 3.020–24.247 | 3.003–24.249 |
| Reflections collected/unique | 5160/4473 | 4920/4532 | 16763/19473 | 9091/7035 | 7937/6427 | 7375/5993 | 8648/7419 |
| [$R_1(I > 2\sigma(I)$); $\omega R_2$(all data)] | $R_1 = 0.0275$, $\omega R_2 = 0.0777$ | $R_1 = 0.0387$, $\omega R_2 = 0.0863$ | $R_1 = 0.0346$, $\omega R_2 = 0.1075$ | $R_1 = 0.0507$, $\omega R_2 = 0.1495$ | $R_1 = 0.0443$, $\omega R_2 = 0.1333$ | $R_1 = 0.0725$, $\omega R_2 = 0.1956$ | $R_1 = 0.0442$, $\omega R_2 = 0.1265$ |
| Goodness-of-fit on $F^2$ | 1.062 | 1.193 | 1.189 | 1.047 | 1.076 | 1.087 | 1.082 |

[a] Standard deviations in parentheses; $R_1 = \sum ||F_o| - |F_c||/\sum |F_o|$; $\omega R_2 = [\sum \omega(F_o^2 - F_c^2)^2/\sum(F_o^2)^2]^{1/2}$.

## 4. Conclusions

Adduct dinuclear and polymer complexes of 4-phpy, pyz, and 4,4′-bpy were obtained by using *cis*-[Rh$_2$$^{II,II}$(4-Me-pf)$_2$(O$_2$CR)$_2$] (R = CF$_3$ and CMe$_3$) as precursor dinuclear units. The dinuclear structures of *cis*-[Rh$_2$$^{II,II}$(4-Me-pf)$_2$(O$_2$CR)$_2$(4-phpy)$_2$] and *cis*-[Rh$_2$$^{II,III}$(4-Me-pf)$_2$(O$_2$CCMe$_3$)$_2$(4-phpy)$_2$]BF$_4$ and polymer structures of [Rh$_2$$^{II,II}$(4-Me-pf)$_2$(O$_2$CR)$_2$(L)]$_n$ (L = pyz and 4,4′-bpy) were confirmed by the X-ray crystal structure analyses. In the adduct complexes, the lantern-type dinuclear core structures with *cis*-(2:2) arrangement of formamidinato (4-Me-pf$^-$) and carboxylato ligands exists without decompositions. For the Rh$_2$$^{II,II}$ adduct dinuclear and polymer complexes, the axial interactions between Rh$_2$$^{II,II}$ core and axial ligands were confirmed by disappearance and blue-shits of the visible bands of the parent dinuclear complexes *cis*-[Rh$_2$$^{II,II}$(4-Me-pf)$_2$(O$_2$CR)$_2$] (R = CF$_3$ and CMe$_3$). The axial interactions make the Rh$_2$$^{II,II}$ dinuclear core oxidized easily, which was observed by the CV measurements for the bis-adduct complexes of *cis*-[Rh$_2$$^{II,II}$(4-Me-pf)$_2$(O$_2$CR)$_2$(4-phpy)$_2$] (R = CF$_3$ and CMe$_3$). The bis-adduct complex *cis*-[Rh$_2$$^{II,III}$(4-Me-pf)$_2$(O$_2$CCMe$_3$)$_2$(4-

phpy)$_2$]BF$_4$ is paramagnetic, which was confirmed by effective magnetic moment value $\mu_{eff}$ = 1.90 $\mu_B$ at 300 K per Rh$_2^{II,III}$ unit (*S* =1/2) with a weak antiferromagnetic interaction among the Rh$_2^{II,III}$ units. The adduct polymer complexes [Rh$_2$(4-Me-pf)$_2$(O$_2$CR)$_2$(L)]$_n$ (L = pyz and 4,4′-bpy) showed Type II gas-adsorption properties for N$_2$. In this study, the dinuclear complex *cis*-[Rh$_2^{II,II}$(4-Me-pf)$_2$(O$_2$CR)$_2$] are shown to provide axial coordination of linker ligands pyz and 4,4′-bpy to result in the formation of adduct polymer complexes and the Rh$_2^{II,II}$ core is oxidized to Rh$_2^{II,III}$ one without decomposition of the lantern-type dinuclear structure was also confirmed by the bis-adduct complexes of 4-phpy. The dinuclear complexes *cis*-[Rh$_2^{II,II}$(4-Me-pf)$_2$(O$_2$CR)$_2$] are considered promising building blocks to be assembled.

**Supplementary Materials:** The following data are available online at https://www.mdpi.com/2312-7481/7/3/39/s1. Rh–Rh bond distance of the related Rh$_2$ complexes (Table S1), selected bond distances and angles of **3–9** (Table S2), absorption spectra of **3** and **4** in CH$_2$Cl$_2$ at the concentration of 1.0 x 10$^{-3}$ M (Figure S1), packing diagram of **5** (Figure S2), and nitrogen adsorption isotherms of **6** (Figure S2), **7** (Figure S3), and **9** (Figure S4).

**Author Contributions:** M.H. conceived and designed the experiment, analyzed the data and wrote the paper. S.N., M.K. and N.Y. performed the experiments. H.A. determined X-ray crystal structures of **5**. M.M. measured diffuse reflectance spectra. H.T. measured nitrogen adsorption isotherms. T.K. helped with the X-ray crystal structure determination. Y.K. determined the X-ray crystal structures of **3**, **4**, and **6–9**. All authors have read and agreed to the published version of the manuscript.

**Funding:** The present work was partially supported by Grants-in-Aid for Scientific Research No. 16K05722 from the Ministry of Education, Culture, Sports, Science and Technology (MEXT), Japan.

**Data Availability Statement:** CCDC-1936749, 1936750, 2059485, 1936751, 1936752, 1936753 and 1936754 contain the supplementary crystallographic data for *cis*-[Rh$_2$(4-Me-pf)$_2$(O$_2$CCF$_3$)$_2$(4-phpy)$_2$] (**3**), *cis*-[Rh$_2$(4-Me-pf)$_2$(O$_2$CCMe$_3$)$_2$(4-phpy)$_2$] (**4**), *cis*-[Rh$_2$(4-Me-pf)$_2$(O$_2$CCMe$_3$)$_2$(4-phpy)$_2$]BF$_4$·2MeOH (**5·2MeOH**), [Rh$_2$(4-Me-pf)$_2$(O$_2$CCF$_3$)$_2$(pyz)]$_n$ (**6**), [Rh$_2$(4-Me-pf)$_2$(O$_2$CCMe$_3$)$_2$(pyz)]$_n$ (**7**), [Rh$_2$(4-Me-pf)$_2$(O$_2$CCF$_3$)$_2$(4,4′-bpy)]$_n$ (**8**), and [Rh$_2$(4-Me-pf)$_2$(O$_2$CCMe$_3$)$_2$(4,4′-bpy)]$_n$ (**9**), respectively. These data can be obtained free of charge from the Cambridge Crystallographic Data Centre via www.ccdc.cam.ac.uk/data_request/cif (accessed on 11 February 2021).

**Acknowledgments:** The authors are grateful to Michiko Egawa (Shimane University) for her measurements of elemental analyses.

**Conflicts of Interest:** The authors declare no conflict of interest.

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
