# Peer review of "Structures and Properties of 4-phpy, pyz, and 4,4′-bpy Adducts of Lantern-Type Dirhodium Complexes with µ-Formamidinato and µ-Carboxylato Bridges"

_magnetochemistry, doi:10.3390/magnetochemistry7030039_

Round 1

Reviewer 1 Report

Manuscript is focused on the synthesis and characterization of  dinuclear structures of cis-[Rh2II,II(4-Me-pf)2(O2CR)2(4-phpy)2] and cis- 648 [Rh2II,III(4-Me-pf)2(O2CCMe3)2(4-phpy)2]BF4 and polymer structures of [Rh2(4-Me- 649 pf)2(O2CR)2(L)]n (L = pyz and 4,4’-bpy). Authors have wisely selected the tools to characterize and these materials.

My concerns are

  1. Authors should discuss why were these characterizations tools used? What properties or applications are authors looking for?
  2. Authors should also point out why are these materials useful?

Author Response

Referee 1

Manuscript is focused on the synthesis and characterization of dinuclear structures of cis- [Rh2II,II(4-Me-pf)2(O2CR)2(4-phpy)2] and cis- 648 [Rh2II,III(4-Me-pf)2(O2CCMe3)2(4- phpy)2]BF4 and polymer structures of [Rh2(4-Me- 649 pf)2(O2CR)2(L)]n (L = pyz and 4,4’-bpy). Authors have wisely selected the tools to characterize and these materials. My concerns are:

Thank you very much for evaluating our work.

  1. Authors should discuss why were these characterizations tools used? What properties or applications are authors looking for?

Thank you very much for the very important comments. CV is good tool to estimate how easily the complexes are oxidized or reduced; which is described at the101st line at the revised manuscript. Magnetic and absorption spectral data are indispensable for looking into spin and electronic states; which is described at 109th line in the revised manuscript. Moreover, molecular structures determined by X-ray single crystal structure analysis is one of the most powerful characterization tool to confirm whether the desired complexes are obtained or not; which is described at 111th line in the revised manuscript. We are looking for assembled complexes showing remarkable magnetic and gas occlusion properties; which is described at 105th line. New materials with such hybridized properties should be important for establishing and developing new research filed of chemistry.

  1. Authors should also point out why are these materials useful?

Thank you very much for the very important comments. The materials mentioned above can be built up by assembling molecules (that is, metal complexes). It is very important to find out building blocks to be assembled. The lantern-type dinuclear Rh2 complexes are promising because both of the axial positions are used for linking the Rh2 complexes in the combination with linkage ligand such as pyrazine (pyz) and 4,4’-bipyridine (4,4’-bpy). Further, Rh2 complexes with the amidinato ligands are robust and easily oxidized without decomposition to become paramagnetic. This is very import as the building block for the hybridized properties we are aiming at; which is described at the 105th and 110th lines in the revised manuscript. We could confirm that Rh2 complexes with the amidinato ligands are promising building blocks, which is described in conclusion (at the 717 – 718th lines).

Reviewer 2 Report

This manuscript describes seven new Rh-based complexes. The manuscript is worth publishing in Magnetochemistry, however, it should be corrected according to the suggestions:
- English should be improved,
-  The 1/X  (Figure 10) should be presented as mol/emu or g/emu.  The description of the magnetic properties of sample 5  should also include the M(H) measurement.
- The visibility of figures presenting the structures should be improved. 

Author Response

Referee 2

This manuscript describes seven new Rh-based complexes. The manuscript is worth publishing in Magnetochemistry, however, it should be corrected according to the suggestions:

Thank you very much for understanding our work.

-English should be improved.

Thank you very much for the important comment. We carefully checked the manuscript and find many grammatical and typing mistakes. We corrected them all we found, which are described in red in the revised manuscript (I awfully mistyped CCDC numbers). I really appreciate the comment.

-The 1/X (Figure 10) should be presented as mol/emu or g/emu. The description of the magnetic properties of sample 5 should also include the M(H) measurement..

Thank you very much for the comments. According to the suggestion, the 1/X was rewritten by emu mol inFugure 10. The M(H) measurement was also performed at 2.0 K. The obtained results are given in Figure 11.

-The visibility of figures presenting the structures should be improved.

Thank you very much for the comment. The resolution of the figures of the structures are increased and the pictures are a little bit enlarged in the revised manuscript.

Reviewer 3 Report

the paper is concerning the characterization of formamidonato rhodium complexes and of interest for researchers involved in organometallic chemistry.

the paper is well organized, but, owing to the peculiar characteristics of the papers published in MDPI journals that is to say the clarity and completeness of the bibliography and a brief review of the literature, to help the reader to enter the specific subject dealt, I would to give some suggestions:

a) addition of the shematic structure of complexes 1 and 2

b) addition in Table 1 of the structural data discussed in the text for comparison

c) a scheme of the energies of the orbitals from DFT calculations

As for cationic Rh(II/III) complexes a better explanation of the shortening of the Rh-Rh distance could be added.

As for the "polymeric" 6-9 complexes some mass data could be useful to have an idea of the n value (polymers or polynuclear compounds). what is the real meaning of the MWs reported in Table 2?

the different extent of N2 adsorption between the complexes 6-9 can be related with the crystal packing?

Author Response

Referee 3

the paper is well organized, but, owing to the peculiar characteristics of the papers published in MDPI journals that is to say the clarity and completeness of the bibliography and a brief review of the literature, to help the reader to enter the specific subject dealt, I would to give some suggestions:

Thank you very much for understanding our work.

    a) addition of the shematic structure of complexes 1 and 2

Thank you very much for the comment. Schematic views of complexes 1 and 2 are given in Scheme 2 in the revised manuscript.

  b)  addition in Table 1 of the structural data discussed in the text for comparison.

Thank you very much for the comment. Actually, so many complexes are used for the comparison. For clarifying the discussion, the Rh-Rh bond distances discussed are listed in Table S1, which is included in supplementary materials.

  c) a scheme of the energies of the orbitals from DFT calculations

Thank you very much for the comment. The calculation results were reported in the ref. 32 (our previous work). The energy diagram is given in ref. 32. To avoid the misunderstanding, the expressions were rewritten at the 186, 187, and 227 – 230th lines in the revised manuscript.

  • As for cationic Rh(II/III) complexes a better explanation of the shortening of the Rh-Rh distance could be added.

Thank you very much for the comment. Explanation “when an electron is removed from the antibonding π* orbital” was added at the 321th line.

  • As for the "polymeric" 6-9 complexes some mass data could be useful to have an idea of the n value (polymers or polynuclear compounds). what is the real meaning of the MWs reported in Table 2?

Thank you very much for the comment. It is impossible to obtain the MS data for the polymer complexes, because the polymer complexes are not soluble in any solvent. If the complexes are dissolved, the complexes could be decomposed to the original dinuclear complexes and likage ligands.

  • the different extent of N2 adsorption between the complexes 6-9 can be related with the crystal packing?

Thank you very much for the comment. Unfortunately, there is no relationship with crystal packing. At the 540 – 544 lines, the void spaces calculated using crystal structure data are described in the revised manuscript. The spaces are considered to collapse in the power powder sample. 

Round 2

Reviewer 3 Report

the manuscript can be accepted in the present form